# Rerouting LLM Routers

**Avital Shafran**
The Hebrew University
Jerusalem, Israel.
avital.shafran@mail.huji.ac.il

**Roei Schuster**
Wild Moose
New York, USA.
roei@wildmoose.ai

**Thomas Ristenpart**
Cornell Tech
New York, USA.
ristenpart@cornell.edu

**Vitaly Shmatikov**
Cornell Tech
New York, USA.
shmat@cs.cornell.edu

## Abstract

LLM routers balance the cost and quality of responding to queries by routing them to a cheaper or more expensive LLM depending on the query's estimated complexity. Routers are a type of what we call "LLM control planes," i.e., systems that orchestrate multiple LLMs.

In this paper, we investigate adversarial robustness of LLM control planes using routers as a concrete example. We formulate LLM control-plane integrity as a distinct problem in AI safety, where the adversary's goal is to control the order or selection of LLMs employed to process users' queries. We then demonstrate that it is possible to generate query-independent "gadget" strings that, when added to any query, cause routers to send this query to a strong LLM. In contrast to conventional adversarial inputs, gadgets change the control flow but preserve or even improve the quality of outputs generated in response to adversarially modified queries.

We show that this attack is successful both in white-box and black-box settings against several open-source and commercial routers. We also show that perplexity-based defenses can be evaded, and investigate alternatives.

## 1 Introduction

Large language models (LLMs) exhibit remarkable capabilities on many tasks. Today, hundreds of open-source and proprietary LLMs are available at different prices, ranging from expensive, state-of-the-art models to cheaper, smaller, less capable ones. Operators of LLMs (especially higher-quality models) typically charge per query, imposing non-trivial costs on LLM-based applications and systems.

Developers who want to integrate LLMs into their applications must therefore consider both utility and cost, i.e., how to maximize the quality of responses while minimizing the cost. The two objectives conflict with each other: larger models tend to generate higher-quality answers but charge more per query. For example, at the time of this writing, GPT-3.5-turbo costs \$0.5/\$1.5 per 1M input/output tokens, GPT-4o-mini \$0.15/\$0.6, GPT-4o \$2.5/\$10, o1-preview \$15/\$60. The difference in quality between models is not uniform across queries. For some queries, even a cheap model can generate an acceptable response. More complex queries require an expensive model to obtain a quality answer.

A natural solution to balancing performance and economic considerations is to take advantage of the availability of multiple LLMs at different price-performance points. Recently proposed **LLM routing** systems (Šakota et al., 2024; Ong et al., 2024; Ding et al., 2024; Martian; Unify) orchestrate two or more LLMs and adaptively route each query to the cheapest LLM they deem likely to generate a response of sufficient quality. In the two-LLM case, let $M_s$ be an expensive, high-quality model and $M_w$ a weaker, lower-grade one. Given query

$q$, the routing algorithm $R(\cdot)$ applies a classifier to $q$ that outputs 0 if $M_w$ is sufficient for answering $q$, or 1 if $M_s$ is required. The system then routes $q$ accordingly.

LLM routing is an example of a general class of systems we call LLM control planes, which orchestrate the use of multiple LLMs to process inputs, as further described in Section 2.

**Our contributions.** First, we introduce ***LLM control plane integrity***, i.e., adversarial robustness of inference *flow* (rather than inference outputs), as a novel problem in AI safety. Robustness of control-plane algorithms to adversarial queries is distinct from adversarial robustness of the LLMs they orchestrate. Attacks on LLMs aim to degrade their outputs. In contrast, control-plane attacks introduced in this paper aim to change the order or selection of LLMs employed by the system while producing the same or even better outputs.

Second, we show that existing LLM routing algorithms can be manipulated by malicious users. We design, implement, and evaluate a method that generates *query-independent* adversarial token sequences which we call "confounder gadgets." If a gadget is added to any query, this query is routed to the strong model with high probability. Next, we show that this attack is effective even in the *transfer* setting where the adversary does not have full knowledge of the target LLM router but has access to another router (e.g., an internally trained surrogate). We also evaluate the integrity of commercial LLM routers, showing that they can be confounded as well.

While our gadgets act like universal adversarial examples against query classifiers used by LLM routers, their objectives are different. Adversarial examples simply aim to change the output of the classifier. Our gadgets need to change the output of the target system's router *and* also ensure that the system's response to the confounded query is the same or better than the response to the original query.

Third, we investigate defenses. Our basic method generates gadgets that have anomalously high perplexity and can be easily filtered out. This defense can be evaded by incorporating a low-perplexity objective into gadget generation. We also discuss higher-level defenses, such as identifying users whose queries are routed to the strong model with abnormal frequency.

Routing attacks can be deployed for various adversarial objectives, e.g., to ensure that the adversary always obtains the highest-quality answer regardless of the target application's internal routing policies and cost constraints, or to maliciously inflate the target's LLM costs. They can affect other users, e.g., by wasting the target's strong-model quota and causing their queries to be routed to the weak model. As LLM control planes grow in importance and sophistication, we hope to motivate further research on their adversarial robustness.

## 2 LLM Control Planes and Routing

Inference using large language models (LLMs) is traditionally monolithic: a single model is applied to an input or sequence of inputs. Today, this methodology can be sub-optimal. State-of-the-art models are often expensive, charging as much as several dollars per API query. Different LLMs may excel at different tasks, and selectively using an appropriate LLM may improve overall quality on a diverse workload. Finally, combining multiple LLMs, even if trained for similar tasks, may become increasingly prevalent as performance improvements of individual LLMs plateaus (Reuters, 2024; The Information, 2024; Bloomberg, 2024).

Researchers and practitioners are developing inference architectures that use multiple LLMs to answer queries. These LLMs are orchestrated by what we call an *LLM control plane* (borrowing the terminology from networking (IBM, 2024)). The control plane may route queries or parts of queries to different LLMs, derive new strings to query to underlying LLMs, combine answers from underlying LLMs, and more.

**LLM routers.** A prominent example of this emerging class of LLM control planes are *LLM routers* (Ding et al., 2024; Ong et al., 2024; Stripelis et al., 2024; Šakota et al., 2024; Lee et al., 2024). LLM routers decide which of the two (or, sometimes, more) LLMs to use to answer a query. In prescriptive routing, the router applies some lightweight classifier to the input

query that determines which underlying LLM to utilize for a response. The classifier is itself a learned function that scores the complexity of the query. Deployments can then configure a score threshold for when to route a query to the more expensive LLM. This threshold can be tuned using representative workloads to achieve a desired cost-performance trade-off. Figure 4 shows the basic workflow of binary LLM routers.

Non-prescriptive routing (Chen et al., 2023; Aggarwal et al., 2023; Yue et al., 2024) picks a response from those produced by multiple LLMs. For example, FrugalGPT (Chen et al., 2023) submits the query to a cascade of models (ordered by price), stopping when it obtains a response classified by the router as sufficient.

In contrast to routers motivated by controlling costs, several LLM router designs focus solely on improving quality of responses (Shnitzer et al., 2023; Narayanan Hari & Thomson, 2023; Feng et al., 2024; Srivatsa et al., 2024).

The LLM routers described thus far do not modify the queries or individual LLM responses. Other types of control planes do. Ensemble approaches such as mixture-of-expert (MoE) (Du et al., 2022; Fedus et al., 2022; Riquelme et al., 2021; Shazeer et al., 2016) architectures select a subset of underlying models to apply to each token of a query and merge their responses. LLM synthesis (Jiang et al., 2023b) architectures operate similarly, but route the entire query to a subset of underlying LLMs and merge their responses. These approaches reduce inference costs by using fewer and/or less complex underlying models.

A key use case for LLM routers is to help LLM-based applications reduce cost. Several commercial routers, including Unify (Unify), Martian (Martian), NotDiamond (NotDiamond), and others, offer this as a service. These services select the optimal LLM and forward the queries, apparently resulting in significant cost savings: up to 98% in the case of Martian (Martian), and $10\times$ in the case of NotDiamond (NotDiamond).

## 2.1 LLM Control Plane Integrity

In this section, we provide a high-level definition of *LLM control plane integrity*. A formal and detailed discussion is provided in Appendix A.

An LLM control plane $R_\omega$ is a potentially randomized algorithm, parameterized by a string $\omega$. It utilizes some number $n$ of LLMs denoted by $\mathcal{M}$. We will mostly focus on the case of $n = 2$, and, for reasons that will be clear in a moment, use $M_{\mathtt{s}}$ ("strong") and $M_{\mathtt{w}}$ ("weak") to denote the underlying LLMs. Given an input $x$, *inference flow* is the sequence of LLM invocations that process $x$. This flow is dictated by an *inference flow policy* that represents the control plane designer's intention regarding the use of the underlying LLMs. For example, in the context of binary LLM routers, the policy might specify that the strong and expensive model is used at most an $\epsilon$ fraction of inferences.

A *control plane integrity adversary* seeks to maliciously guide the inference flow by overriding the intended inference flow policy for obtaining some adversarial goal. Informally, control plane integrity means that decisions made by control plane algorithms cannot be subverted by adversarial queries. We focus on predictive LLM routing as a concrete example of a control plane used in real-world LLM-based systems to manage inference cost.

**Threat models.** An adversary may seek to *inflate the costs* of a victim application that utilizes an LLM control plane for cost management. Another adversarial goal is *arbitrage*. Consider an application that charges $X$ dollars per query, whereas directly using $M_{\mathtt{s}}$ costs $Y > X$. This makes economic sense if the bulk of queries are routed internally to a cheaper model $M_{\mathtt{w}}$. An input adaptation attack in this setting can gain (indirect) access to $M_{\mathtt{s}}$, obtaining an arbitrage advantage of $Y - X$ per query. To be effective, adaptations should not lower response quality (i.e., extract all value out of rerouting to $M_{\mathtt{s}}$).

A lot of prior research on adversarial ML focused on attacks like jailbreaking and poisoning that impact the end users of LLMs. In our setting, the primary victims are applications that employ vulnerable LLM control planes. Furthermore, their users will be affected, too, if their queries are unfairly routed to the weak model. The adversary, on the other hand, gains unfair access to the high-quality LLM.

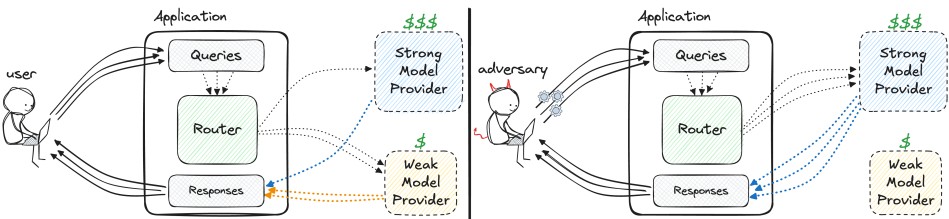

Figure 1: Overview of our attack on LLM control plane integrity. The attack adds to each query a prefix (represented by the gear), called a "confounder gadget," that causes the router to send the query to the strong model.

We assume that the victim application's prompt includes a substring that can be controlled by the adversary. For example, chatbots, coding assistants, writing assistants, and other LLM-based applications include user inputs into their prompt. We consider several levels of knowledge that an adversary may have about the victim application's internal router.

In the *white-box* case, the adversary knows the control plane algorithm and its parameters $\omega$. In the *black-box (transfer)* case, the adversary does not know the control plane algorithm $R$ and $\omega$ for the target model, but knows instead another control plane algorithm $R'_{\omega'}$ and its parameters. We refer to $R'_{\omega'}$ as the *surrogate*. For example, this could arise if an adversary trains their own router. In this setting our attacks are also *zero-shot* in that they do not require any prior interaction with the target control plane.

## 3 Confounding Control Planes with Gadgets

Control plane attacks modify queries to mislead or "confound" the routing logic into routing these queries to an LLM of the adversary's choosing. We focus on the binary router setting in which the router applies a learned parameterized scoring function $S_\theta(\cdot)$, where $\theta$ refers to the parameters, and routes any query whose score exceeds some threshold $\tau$ to the strong LLM $M_\mathbf{s}$. This setting has been the focus of (Ong et al., 2024; Ding et al., 2024; Lee et al., 2024) and is deployed in practice (see Section 6). We provide a more formal definition in Appendix B. See Figure 1 for a depiction of our attack setting.

**Confounder gadgets.** Our approach works as follows. Given a query $x_i$, we prepend a *confounder gadget* $c_i$, which is a short sequence of adversarially chosen tokens. The modified query is $\hat{x}_i = c_i \| x_i$ where $\|$ denotes string concatenation. Intuitively, we will use optimization to search for confounders that trick the scoring function into ranking $\hat{x}_i$ as sufficiently complex to require the strong model.

In the white-box, query-specific setting, we can choose $c_i$ as a function of $x_i$ and the known parameters $\omega = (S, \theta, \tau)$. To avoid searching a confounder for every query, we focus instead on the *query-independent* setting, where a single confounder can be prepended to all queries.

We begin by fixing a confounder length of $n$ tokens and let $\mathcal{I}$ be a token dictionary (it should be a sufficiently large subset of the token dictionary used by $S$). Then we set the gadget to initially be $n$ tokens all fixed to the same value from $\mathcal{I}$. The exact choice of the initialization token is not important; in our implementation, we used the first token in the dictionary ('!'). Denote this initial confounder as $c^{(0)} = [c_1^{(0)}, c_2^{(0)}, \ldots, c_n^{(0)}]$.

Then, we perform a hill-climbing style approach to find a good confounder. For each iteration $t \in [T]$, where $T$ is the total number of iterations, do the following:

(1) Select a target index $j \in [1, n]$ uniformly.

(2) Generate a set $\mathcal{B}$ of $B + 1$ candidates. First set $\tilde{c}_0 = c^{(t)}$, the current confounder. To generate $B$ additional candidates, select replacement tokens from $\mathcal{I}$ uniformly, forming

the set $\{t_b \leftarrow \mathcal{I}\}_{b=1}^B$. Replace the $j^{\text{th}}$ token in the current confounder $\tilde{c}_0$ with $t_b$, forming $\tilde{c}_b = [c_1^{(t)}, \ldots, c_{j-1}^{(t)}, t_b, c_{j+1}^{(t)}, \ldots, c_n^{(t)}]$. Let $\mathcal{B} = \{\tilde{c}_0, \ldots, \tilde{c}_B\}$.

(3)  Find the candidate that maximizes the score: $c^{(t+1)} \leftarrow \arg\max_{c \in \mathcal{B}} S_\theta(c)$.

The final confounder $c^{(T)}$ can be used with any query $x_i$ by prepending it to obtain $\hat{x}_i = c \| x_i$. We early abort if, after 25 iterations, there is no update to the confounder gadget. Technically, we could abort early if we find a confounder whose score exceeds $\tau$. Running further can be useful when an adversary does not know $\tau$.

The attack's runtime is dominated by $T \cdot B$ times the cost of executing $S$. In practice, $S$ are designed to be fast (otherwise routers would significantly increase the latency of applications that use them). We report precise timings later; in summary, the attack is fast because we can set $T$ to be relatively small and still find high-scoring confounders.

Due to the randomness in index and token selection, the method converges to different, yet similarly effective, confounder gadgets on each run. Our evaluation will thus measure average performance over multiple gadgets.

**The black-box setting: confounders that transfer.** Finally, the attacks so far are in the white-box setting, where the attacker can optimize directly against $S_\theta$. While in some cases routing control planes will be public knowledge, in others, including the proprietary control planes we explore in Section 6, they are hidden. This gives rise to the black-box setting. While an attacker might seek to perform model extraction attacks (Tramèr et al., 2016; Lowd & Meek, 2005) to learn $\theta$, we instead explore attacks that transfer from one router to another.

We assume the adversary has access to a *surrogate* router $R'_{\omega'}$ that is trained on data similar to that used for the target router. Then the attack is the same as above, except that we use the surrogate's scoring function $S'_{\theta'}$ instead of the target's $S_\theta$. Again, we will see that query-independent confounders found for the surrogate transfer well to other routers.

We have experimented with variations of this approach that don't work quite as well, for example adding $c$ as a suffix instead of a prefix. See Appendix I for details.

## 4  Open-Source Routers: Experimental Setup

This section explains our experimental setup (summarized in Figure 3). We provide more details in Appendix D. In all experiments, we assume that the adversary's goal is to reroute queries to the strong model. In Appendix L, we evaluate rerouting to the weak model.

**Target routers.** We evaluate the four prescriptive routing algorithms from Ong et al. (2024), which provides open-source code and trained parameters for a representative variety of approaches: similarity-based classification (Stripelis et al., 2024; Lee et al., 2024), an MLP constructed via matrix factorization (Stripelis et al., 2024), BERT-based classification (Ding et al., 2024; Stripelis et al., 2024; Šakota et al., 2024), and a fine-tuned LLM.

**Underlying LLMs.** Ong et al. (2024) trained the routers with GPT-4-1106-preview (Achiam et al., 2023) as the strong model and Mixtral 8x7B (Jiang et al., 2024) as the weak model. They report successful generalization between the underlying LLMs, stating that their routers trained for a particular strong-weak LLM pair can be used with other pairs as well.

To allow our evaluation to scale, we use as the strong model $M_s$ the open-sourced Llama-3.1-8B (Meta, 2024b) and as $M_w$ the 4-bit quantized version of Mixtral 8x7B (for efficiency reasons). This reduced the cost of our experiments by avoiding expensive GPT API calls and lowering the computational costs of Mixtral. Unless mentioned otherwise, all of our results are for this router pair (LLM Pair 1). In Appendix M.2, we perform less extensive experiments with the original strong/weak model pair (LLM pair 4). We additionally performed experiments for the case where the weak model produces much worse responses than the strong model—see Appendix M.

| | MT-Bench | | MMLU | | GSM8K | |
|---|---|---|---|---|---|---|
| | Upgrade | Strong | Upgrade | Strong | Upgrade | Strong |
| $R_{SW}$ | $100 \pm 0$ | $81 \rightarrow 100 \pm 0$ | $90 \pm 1$ | $43 \rightarrow 94 \pm 1$ | $98 \pm 0$ | $52 \rightarrow 99 \pm 0$ |
| $R_{MF}$ | $100 \pm 0$ | $58 \rightarrow 100 \pm 0$ | $78 \pm 4$ | $53 \rightarrow 90 \pm 2$ | $100 \pm 0$ | $54 \rightarrow 100 \pm 0$ |
| $R_{CLS}$ | $100 \pm 0$ | $67 \rightarrow 100 \pm 0$ | $100 \pm 0$ | $47 \rightarrow 100 \pm 0$ | $100 \pm 0$ | $56 \rightarrow 100 \pm 0$ |
| $R_{LLM}$ | $73 \pm 5$ | $57 \rightarrow 88 \pm 2$ | $95 \pm 1$ | $53 \rightarrow 98 \pm 1$ | $94 \pm 3$ | $53 \rightarrow 97 \pm 1$ |

Table 1: The white-box attack's rerouting success rate. "Upgrade" is the percentage of "Weak" queries successfully rerouted to the strong model by adding a confounder gadget. "Strong" is the change in the percentage of queries that are routed to the strong LLM $M_s$.

**Evaluation datasets.** We will evaluate our attacks using three standard LLM benchmarks as workloads: MT-Bench (Zheng et al., 2023), a dataset of 160 open-ended questions, MMLU (Hendrycks et al., 2021), a dataset of 14,042 multi-choice questions, and GSM8K (Cobbe et al., 2021), a dataset of 1,319 grade-school math problems. We provide more details about benchmark evaluation in Appendix D.

**Router calibration.** For each workload, we must calibrate each router by setting the threshold $\tau$ to achieve some target fraction $\epsilon$ of queries routed to the strong model. For our experiments here, we set $\epsilon = 0.5$, meaning the goal is to have about half the queries routed to the strong model. We provide more details about calibration in Appendix D and evaluate other values of $\epsilon$ in Appendix E.

## 5    Rerouting Open-Source Routers

Unless otherwise specified, our evaluation focuses on the setting where the attacker first finds a fixed, query-independent set of gadgets and uses them to attack all queries. In Appendix N, we evaluate query-specific gadgets which are more effective but also more expensive to generate. In Appendix J, we evaluate optimization-free alternatives for generating confounder gadgets. We discuss defense mechanisms in Section 7.

**White-box confounder gadget generation.** Following our attack framework described in Section 3, we construct a query-independent control-plane gadget designed to confuse each router. For the white-box setting, we set batch size $B = 32$ and iteration number $T = 100$, ignoring thresholds. We generate four sets of $n = 10$ gadgets per router (see Appendix C for examples). We discuss runtime and optimization convergence in Appendix F.

When reporting the scores below, we average over $n$ gadgets used with all 72 MT-bench queries, and sets of 100 randomly selected queries from MMLU and GSM8K. None of these test queries were used in the training of the routers or their calibration.

**Rerouting success rates.** We measure the percentage of *upgraded* queries, i.e., queries that were originally routed to the weak model and after being modified with the confounder gadget, were rerouted to the strong model. Table 1 shows that our attack successfully reroutes almost all weak queries to the strong model. *No* queries are "downgraded," i.e., rerouted from the strong to weak model. We also show the increase in the percentage of queries routed to the strong model before and after modifying queries with our gadgets.

**Quality of responses.** A successful gadget must reroute queries to the strong model while preserving or improving quality of responses to the *modified* query. We use perplexity and benchmark-specific scores to evaluate response quality. Table 2 shows that for LLM pair 1, average response perplexity does not significantly change. To the extent that it does, it usually somewhat decreases, indicating more "natural" responses. Table 3 shows that responses to confounded queries' average scores (1-10 quality scores for MT-bench, % correct for MMLU/GSM8K) are typically no worse (sometimes better) than responses to the original queries. We provide full details and further discussion in Appendix G.

| | MT-Bench | | MMLU | | GSM8K | |
|---|---|---|---|---|---|---|
| | Original | Confounded | Original | Confounded | Original | Confounded |
| $R_{SW}$ | 13.8 | $12.3 \pm 0.2$ | 20.4 | $20.1 \pm 0.1$ | 17.1 | $15.1 \pm 0.3$ |
| $R_{MF}$ | 12.6 | $12.3 \pm 0.2$ | 20.0 | $20.3 \pm 0.1$ | 17.0 | $15.2 \pm 0.3$ |
| $R_{CLS}$ | 13.1 | $12.1 \pm 0.2$ | 20.2 | $20.5 \pm 0.1$ | 17.0 | $15.0 \pm 0.2$ |
| $R_{LLM}$ | 12.7 | $12.7 \pm 0.4$ | 21.0 | $19.6 \pm 0.1$ | 16.4 | $15.2 \pm 0.3$ |

Table 2: Average perplexity of responses to the original and confounded queries, in the white-box setting for LLM pair 1.

| | MT-Bench | | MMLU | | GSM8K | |
|---|---|---|---|---|---|---|
| | Original | Confounded | Original | Confounded | Original | Confounded |
| $R_{SW}$ | 8.4 | $8.3 \pm 0.0$ | 61 | $66 \pm 0$ | 46 | $64 \pm 1$ |
| $R_{MF}$ | 8.4 | $8.4 \pm 0.0$ | 64 | $64 \pm 1$ | 50 | $67 \pm 1$ |
| $R_{CLS}$ | 8.4 | $8.3 \pm 0.0$ | 63 | $65 \pm 0$ | 50 | $63 \pm 1$ |
| $R_{LLM}$ | 8.3 | $8.2 \pm 0.1$ | 67 | $66 \pm 0$ | 44 | $64 \pm 1$ |

Table 3: Average benchmark-specific scores of responses to the original and confounded queries, in the white-box setting for LLM pair 1.

**Black-box attack results.** Consider an attacker that does not know the algorithm used by the target router but has access to a surrogate router to mount a transferability-based attack. We evaluate all combinations of surrogate and target routers. Upgrade rates are lower than in the white-box setting, with an average of 80%, 66%, 71%, and 62% for $R_{SW}$, $R_{MF}$, $R_{CLS}$, and $R_{LLM}$ respectively, but still high, i.e., the attack transfers. The LLM-based algorithm $R_{LLM}$ has the lowest rates, perhaps because it is the most complex of the four. The downgrade rate is 1.2% on average and 0% in most cases.

We additionally evaluate the response quality and show that as in the white-box setting, the attack does not increase the average perplexity for LLM pair 1 nor does it decrease the average benchmark-specific scores, in most cases. We provide full results in Appendix H.

## 6 Rerouting Commercial Routers

We evaluate our rerouting attack on several commercial routers: Unify (Unify), NotDiamond (NotDiamond), OpenRouter (OpenRouter), and Martian (Martian). As these are only available via black-box APIs, we use our black-box attack with the 40 gadgets optimized for the open-sourced routers $R_{SW}$, $R_{MF}$, $R_{CLS}$, and $R_{LLM}$ (10 per router). We evaluate using the MT-bench benchmark.

**Unify.** This router lets users specify a list of models from different providers and a metric configuration for routing decisions. The available metrics are quality, time to first token, inter-token latency, and cost. We configure the router to choose between GPT-4o (OpenAI, 2024b) as the strong model and Mixtral 8x7B as the weak model. We manually calibrate the router such that 49% of the original, unmodified queries being routed to the strong model and 51% to the weak model, resulting in a total cost of $0.13 for the 72 MT-bench queries.

Adding confounder gadgets generated for the four open-source routers results in upgrade rates of 79%, 88%, 91%, and 89%, respectively, averaged across 10 gadgets. Downgrade rates are all zero. Average cost across the 10 gadgets increased to $0.22, $0.23, $0.22, and $0.21, respectively reflecting an average cost increase by a factor of $1.7\times$.

**NotDiamond.** This router lets users route their queries to a list of predefined models. Available objectives are to maximize quality, or balance quality and cost, or balance quality and latency. Exact details of routing logic are not specified. We focus on "cost-aware" routing, for which the docs state "NotDiamond will automatically determine when a query is simple enough to use a cheaper model without degrading the quality of the response."

Again we use GPT-4o (strong) and Mixtral-8x7b (weak). The router sends 18% of the original queries to the weak model. Our gadgets for $R_{SW}, R_{MF}, R_{CLS}$, and $R_{LLM}$ upgrade the LLM at rates of $21\%, 18\%, 21\%$, and $15\%$, respectively. Downgrade rates are 1–3%.

To begin with, NotDiamond aggressively routes to the stronger model even for unmodified queries, which may be the reason the attack's success rate is not as high as in other settings. We explored using NotDiamond with various other LLM pairs, and observed similarly unbalanced routing and similar upgrade and downgrade rates.

**OpenRouter.** This API offers routing between Llama-3-70b (weakest, cheapest), Claude-3.5-Sonnet (middle option), and GPT-4o (strongest, most expensive). Queries are routed "depending on their size, subject, and complexity."[1]

With OpenRouter, 96% of the original queries are routed to the weaker Llama, 4% to the stronger GPT, and none to Claude. The total cost for all original queries is $0.03. After adding confounder gadgets, queries originally routed to Llama (96% of queries) present an interesting pattern: for some gadgets, *all* of those queries are rerouted to GPT, and for other gadgets, there is no impact whatsoever. The universally effective gadgets are generated at roughly similar rates across most of the attacker's surrogate routers (20%-40% of gadgets generated using $R_{SW}, R_{MF}, R_{LLM}$, 0% using $R_{CLS}$). Using any of the universally effective gadgets increases the overall cost by $8\times$, to an average of $0.25.

**Martian.** This router is supposed to let the user provide a list of models and to specify the maximum amount the user is willing to pay for a query or for 1M tokens. Unfortunately, as of November, 2024, the router appears to forward queries to the same LLM regardless of the user's list. We notified Martian about this behavior and excluded it from our evaluation.

# 7  Defenses

**Perplexity-based filtering.** Perplexity is a measure of how "natural" the text looks (see 5), and it can be used to detect adversarial manipulation (Jain et al., 2023; Alon & Kamfonas, 2023). In this defense, we assume the defender has access to a set of unmodified queries. The defender computes their perplexity to establish a baseline threshold; new queries whose perplexity exceed the threshold (i.e. are abnormally "unnatural") are flagged as adversarial.

To evaluate the efficacy of this defense, we compare the perplexity values of the original GSM8K queries and queries confounded with one randomly selected gadget produced using $R_{SW}$. Perplexity is calculated using GPT-2. Figure 2 presents histograms of perplexity values of original vs. confounded queries, and the ROC curve for the defense that uses a perplexity threshold to distinguish between original and confounded queries. These results indicate that confounded queries are readily distinguishable from the original queries using perplexity values (ROCAUC≈1). Results are similar for other gadgets and benchmarks, which we omitted due to space constraints.

Unfortunately, this defense can be evaded if an adversary incorporates a perplexity constraint into the gadget generation process, such that it maximizes the score of the routing algorithm $R$ and simultaneously aligns the gadget's perplexity to some predefined value. We find that this adaptive attack attains comparable success rates to the original attack, but renders perplexity values of confounded queries similar to those of original queries, with a distinguisher's ROCAUC values of 0.5-0.7. Appendix O provides full details of this adaptive attack and its evaluation.

**LLM-based filtering.** Even though adversarially modified queries cannot be easily detected using perplexity, they may still be "unnatural." A possible defense is to employ an oracle LLM to determine if the query is natural or not. This defense requires the router to invoke an additional LLM for every processed query, which is computationally expensive. In fact, this would undermine the very purpose of routing, which is to save costs (Section 1).

---

[1] https://openrouter.ai/openrouter/auto

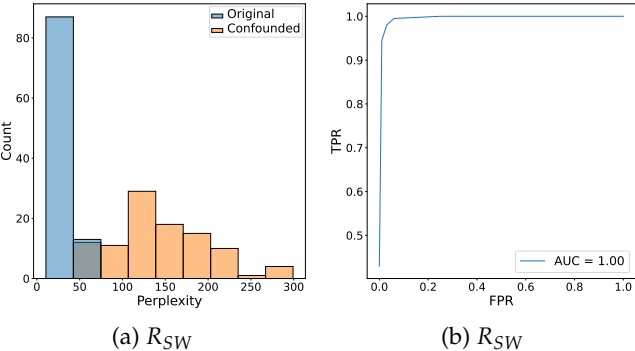

(a) $R_{SW}$          (b) $R_{SW}$

Figure 2: Perplexity of the original queries in the GSM8K benchmark compared to the perplexity of confounded queries using a single uniformly sampled gadget. We additionally present the ROC curve of the defense that detects confounded queries by checking if they cross a perplexity threshold, and it's corresponding ROCAUC score. Confounded queries have significantly higher perplexity values, and are thus easy to recognize and filter out.

| | MT-Bench | | | MMLU | | | GSM8K | | |
| | $N = 1$ | $N = 2$ | $N = 3$ | $N = 1$ | $N = 2$ | $N = 3$ | $N = 1$ | $N = 2$ | $N = 3$ |
|---|---|---|---|---|---|---|---|---|---|
| $R_{SW}$ | $80 \pm 4$ | $96 \pm 2$ | $91 \pm 3$ | $68 \pm 4$ | $85 \pm 2$ | $75 \pm 5$ | $84 \pm 3$ | $97 \pm 1$ | $95 \pm 2$ |
| $R_{MF}$ | $81 \pm 5$ | $99 \pm 1$ | $97 \pm 1$ | $66 \pm 3$ | $86 \pm 1$ | $82 \pm 3$ | $86 \pm 2$ | $98 \pm 0$ | $96 \pm 1$ |
| $R_{CLS}$ | $78 \pm 5$ | $96 \pm 2$ | $89 \pm 4$ | $62 \pm 5$ | $84 \pm 3$ | $69 \pm 5$ | $76 \pm 3$ | $91 \pm 2$ | $83 \pm 5$ |
| $R_{LLM}$ | $70 \pm 3$ | $92 \pm 2$ | $98 \pm 4$ | $52 \pm 4$ | $65 \pm 3$ | $48 \pm 4$ | $74 \pm 4$ | $87 \pm 3$ | $78 \pm 6$ |

Table 4: Average upgrade rate when evaluating against $N + 1$ routers, including the router used for optimization (represented by the left column). For a given value of $N$ all possible combinations were evaluated and averaged. Attack effectiveness decreases in comparison to the standard white-box setting in Table 1, yet a significant upgrade rate persists.

Furthermore, it is possible to optimize gadgets so that they both have low perplexity and appear "natural" to LLM evaluators (Zhang et al., 2024a).

**Paraphrasing.** We can consider an "active" defense that paraphrases queries using an oracle LLM before dispatching them to the router. The paraphrased query might not contain the original gadget, and might not confound the router. This defense is likely impractical. First, as with LLM-based filtering, it requires an expensive LLM invocation for each query. Second, it may degrade the quality of LLM responses, which are sensitive to phrasing.

**Detecting anomalous user workloads.** Another possible defense requires the router to monitor individual user workloads, identify users whose queries are routed to the strongest model with an abnormally high frequency, and impose user-specific thresholds dynamically calibrated to route a consistent fraction of queries to the strong model. Such user-specific routing would complicate implementations and make inaccurate decisions for a user until there is sufficient data about their queries. The defense could still be circumvented in settings where attackers can create a large number of Sybil users.

**Multiple routers.** Another possible defense can utilize multiple routers by making the routing decision based on the majority vote of these routers. We evaluate this defense in the white-box setting, where we use gadgets optimized against a single router and the routing decision is based on the router used for optimization and $N$ other routers (where $N$ ranges from 1 to 3, given 4 total routers), testing all combinations. Table 4 shows that this method decreases attack effectiveness, yet a significant upgrade (i.e., successful attack) rate persists.

As a possible countermeasure, we evaluate the effect of optimizing the gadgets against $M$ routers. The optimization objective in this case is to maximize the average score, with $M$

| MT-Bench | | | MMLU | | | GSM8K | | |
|---|---|---|---|---|---|---|---|---|
| $M=2$ | $M=3$ | $M=4$ | $M=2$ | $M=3$ | $M=4$ | $M=2$ | $M=3$ | $M=4$ |
| $85 \pm 2$ | $100 \pm 0$ | $100 \pm 0$ | $74 \pm 1$ | $91 \pm 1$ | $88 \pm 2$ | $90 \pm 1$ | $99 \pm 0$ | $99 \pm 0$ |

Table 5: Average upgrade rate when optimizing against $M > 1$ routers and evaluating using $N = M$ routers. For a given value of $M$ all possible combinations were evaluated and averaged. This degrades the performance of the multi-router defense although at the cost of slower gadget generation

ranging from 2 to 4. Table 5 indicates that optimizing against multiple routers degrades the performance of using $N = M$ routers as a defense mechanism, although at the cost of slower gadget generation.

## 8 Related Work

There is a large body of research on adversarial examples Goodfellow et al. (2015); Papernot et al. (2016; 2017). For example, HotFlip Ebrahimi et al. (2017) is an early attack against text classifiers that employed token-swapping optimization similar to ours. More recent *prompt injection* adversarial examples specifically target LLMs to extract information or bypass safety guardrails Liu et al. (2023); Schulhoff et al. (2023); Zou et al. (2023); Wei et al. (2023); Zhu et al. (2023); Chu et al. (2024). As explained in Section 1, the purpose of adversarial examples in our case is to change the control flow while maintaining or improving the quality of the system's responses. See Appendix P for a survey of other attacks against LLM-based systems.

## 9 Conclusion

LLM routers balance quality and cost of inference by routing different queries to different LLMs. They are an example of a broader, emerging class of systems we call "LLM control planes" that orchestrate multiple LLMs to respond to queries.

We introduced and defined a new safety property, LLM control plane integrity. Informally, this property holds if an adversarial user cannot influence the routing decisions made by the control plane. To show that existing LLM routers do not satisfy this property, we designed, implemented, and evaluated a black-box optimization method for generating query-independent "confounder gadgets." When added to any query, the gadget confuses the router into routing the query to the adversary-chosen LLM, yet preserves the quality of that LLM's response to the query.

We demonstrated the efficacy of confounder gadgets on multiple open-source and commercial LLM routers, discussed defenses, and indicated directions for future research.

## Acknowledgments

This research was supported in part by the Google Cyber NYC Institutional Research Program, the Israel Science Foundation (Grant No. 1336/22), and the European Union (ERC, FTRC, 101043243). Views and opinions expressed are however those of the author(s) only and do not necessarily reflect those of the European Union or the European Research Council. Neither the European Union nor the granting authority can be held responsible for them.

## Ethics Statement

As LLMs approach the limitations of scaling laws, researchers and practitioners are adopting inference architectures that leverage multiple LLMs to response to user queries. It is important to proactively identify potential security vulnerabilities in these architectures.

Our work initiates the study of integrity vulnerabilities in multi-LLM orchestrations, how they can be exploited, and how these attacks can be mitigated. The purpose of this work is to motivate research on protecting control-flow integrity in multi-LLM systems and help the community design more robust and trustworthy multi-LLM systems.

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

| Routers | Notation |
|---|---|
| Similarity-weighted ranking | $R_{SW}$ |
| Matrix factorization | $R_{MF}$ |
| BERT classifier | $R_{CLS}$ |
| LLM scoring | $R_{LLM}$ |

| LLM pair | Strong ($M_s$) | Weak ($M_w$) |
|---|---|---|
| 1 | Llama-3.1-8B | 4-bit Mixtral 8x7B |
| 2 | Llama-3.1-8B | Mistral-7B-Instruct-v0.3 |
| 3 | Llama-3.1-8B | Llama-2-7B-chat-hf |
| 4 | GPT-4-1106-preview | 4-bit Mixtral 8x7B |

| Benchmark | Description |
|---|---|
| MT-Bench (Zheng et al., 2023) | 160 open-ended questions |
| MMLU (Hendrycks et al., 2021) | 14,042 multi-choice questions |
| GSM8K (Cobbe et al., 2021) | 1,319 grade-school math problems |

Figure 3: Summary of our setup for routers, underlying LLMs, and benchmark datasets used in the experiments.

## A  LLM Control Plane Integrity

In Section 2.1 we provided a high-level definition of LLM control planes, inference flows, and the control plane integrity adversary. In this section we provide a more formal and extensive discussion of these definitions.

**Formalizing control planes.** An LLM control plane $R_\omega$ is a potentially randomized algorithm, parameterized by a string $\omega$ called the parameters. It utilizes some number $n$ of LLMs denoted by $\mathcal{M}$. We will mostly focus on the case of $n = 2$, and, for reasons that will be clear in a moment, use $M_s$ ("strong") and $M_w$ ("weak") to denote the underlying LLMs. Then inference on an input $x \in \mathcal{X}$ for some set $\mathcal{X}$ of allowed queries is performed by computing $y \leftarrow_\$ R_\omega^{\mathcal{M}}(x)$. Here we use $\leftarrow_\$$ to denote running $R$ with fresh random coins; we use $\leftarrow$ when $R$ is deterministic. We focus on inference for a single query, but it is straightforward to extend our abstraction to sessions: the controller would maintain state across invocations, potentially adapting its behavior as a function of a sequence of queries and responses.

LLM control planes should, in general, be computationally lightweight, at least compared to the underlying LLMs. When a control plane is deployed to reduce inference costs by using a cheaper LLM for some queries, an expensive control plane would eat into the savings. For example, predictive binary LLM routers use relatively simple classifiers to determine whether to use $M_s$ or $M_w$ to respond to a query.

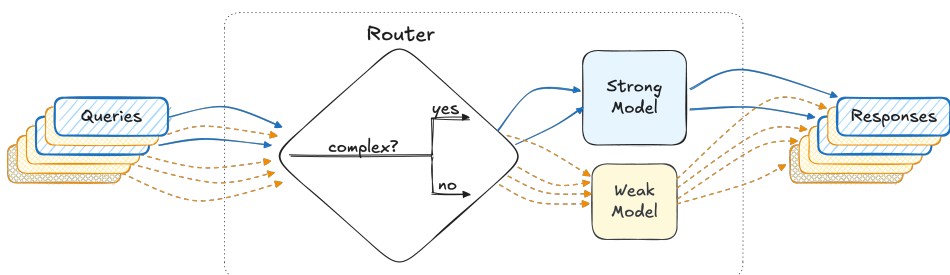

Figure 4: LLM routers classify queries and route complex ones to an expensive/strong model, others to a cheaper/weak model.

**Inference flow.** Given a set of LLMs $\mathcal{M}$, a control plane $R_\omega$, and an input $x$, an LLM inference flow is the sequence of LLM invocations $M_{i_j}(z_j)$ for $1 \le j \le m$ and $i_j \in \{w, s\}$ made when executing $R_\omega^{\mathcal{M}}(x)$. Here $m$ is the total number of invocations, and $z_1, \ldots, z_m$ are the queries. Should $R$ be randomized, the sequence and its length are random variables. An

inference flow can be written as a transcript

$$T = (i_1, z_1), (i_2, z_2), \ldots, (i_m, z_m)$$

of pairs of model indexes $i_j \in \{\mathtt{w}, \mathtt{s}\}$ and model inputs $z_j$. For simplicity, we assume that execution proceeds serially. For binary routers, we have $m = 1$ and $T \in \{(\mathtt{w}, x), (\mathtt{s}, x)\}$. We write submitting a sequence of inferences $\vec{x} = \vec{x}_1, \ldots, \vec{x}_q$ to a control plane as

$$R_\omega^{\mathcal{M}}(\vec{x}) = (R_\omega^{\mathcal{M}}(\vec{x}_1), \ldots, R_\omega^{\mathcal{M}}(\vec{x}_q))$$

In the binary router case, each inference results in a single LLM invocation, but in general inference could involve multiple LLM invocations.

An *inference flow policy* dictates the control plane designer's intention regarding use of the underlying models. For example, an application may want to ensure that only a small fraction of queries go to the expensive model $M_\mathtt{s}$. We can define this as a predicate over a sequence of transcripts. In our binary router example, the policy can be more simply defined as a predicate $\mathcal{P}$ over (input, model) pairs $(\vec{x}_1, i_1), \ldots, (\vec{x}_q, i_q)$ since this fully defines the sequence of transcripts. For example, a policy might specify that the strong model is used in at most an $\epsilon$ fraction of inferences:

$$\mathcal{P}((\vec{x}_1, i_1), \ldots, (\vec{x}_q, i_q)) = \left( \sum_{j=1}^{q} \frac{\mathbb{I}(i_j)}{q} \leq \epsilon \right)$$

where $\mathbb{I}(i_j) = 1$ if $i_j = \mathtt{s}$ and $\mathbb{I}(i_j) = 0$ if $i_j = \mathtt{w}$. In other words, the predicate is that the fraction of queries routed to the strong model is bounded by $\epsilon$.

**Control plane integrity.** A *control plane integrity adversary* is a randomized algorithm $\mathcal{A}$ that seeks to maliciously guide inference flow.

In an unconstrained LLM control plane integrity attack, the adversary $\mathcal{A}$ seeks to generate inputs $\vec{x} = \vec{x}_1, \ldots, \vec{x}_q$ such that running $R_\omega^{\mathcal{M}}(\vec{x})$ generates a transcript for which $\mathcal{P}((x_1, i_1), \ldots, (x_q, i_q)) = 0$. This attack could be launched by an adversary who wants to maximize inference costs for a victim application using an LLM router.

A harder setting requires input adaptation, where the adversary is given inputs $x_1, \ldots, x_q$ and it must find new inputs $\hat{x}_1, \ldots, \hat{x}_q$ for which the transcript resulting from $\mathcal{P}((\hat{x}_1, i_1), \ldots, (\hat{x}_q, i_q)) = 0$. There will be some competing constraint, such as that $x_j$ and $\hat{x}_j$ are very similar for each $j$, or that the outputs $y_j \leftarrow^\$ R_\omega^{\mathcal{M}}(x_j)$ and $\hat{y}_j \leftarrow^\$ R_\omega^{\mathcal{M}}(\hat{x}_j)$ are close. In the routing context, the adversary's goal is to increase the fraction of queries that get routed to the strong model, in order to improve the overall quality of responses, drive up the victim application's inference costs, or both.

**Relationship to adversarial examples.** Evasion attacks (Dalvi et al., 2004; Lowd & Meek, 2005; Szegedy et al., 2013) against inference systems, aka adversarial examples (Goodfellow et al., 2015; Papernot et al., 2016; 2017), would, in our setting, seek to find a small modification $\Delta$ to an input $x$ such that $R_\omega^{\mathcal{M}}(x + \Delta) \neq R_\omega^{\mathcal{M}}(x)$ where addition is appropriately defined based on input type (e.g., slight changes to query text).

Our attack setting is not the same because in our case, the adversary's goal is not just to mislead the classifier. Instead, the goal is defined with respect to the overall system behavior, e.g., to drive up the victim's inference cost or unfairly improve the quality of outputs. Attacking the control-plane component with adversarial inputs is but a means towards these goals.

Simply changing routing decisions with adversarial examples (i.e., an unconstrained attack on control plane integrity) is not enough. In the input adaptation attack, the adversary seeks to modify the inference flow yet *not* change the responses of the strong underlying LLM to the extent possible. **Adversarial inputs against the control plane must preserve or even improve the outputs generated by the overall system**.

In the other direction, undermining LLM control plane integrity could facilitate evasion attacks. For example, if $R_\omega^{\mathcal{M}}$ is used to classify malicious content by combining LLMs tuned

to different types of harm categories, then modifying inputs to force inference flows away from appropriate models could aid evasion. We leave this to future work.

## B  LLM Routers

In Section 3 we discuss our methodology for attacking LLM control plane integrity. We focus on the binary router setting in which the router applies a learned scoring function to input queries and routes any query whose score exceeds some threshold $\tau$ to the strong LLM $M_{\mathtt{s}}$. In this section we provide a more formal description of LLM routers.

We consider a router $R_\omega^{\mathcal{M}}$ for $\mathcal{M} = \{M_{\mathtt{w}}, M_{\mathtt{s}}\}$, where $\omega$ consists of a scoring function $S$, scoring function's parameters $\theta$, and a threshold $\tau \in \mathbb{R}^+$. For notational brevity we just write $R_\omega$ below, with $\mathcal{M}$ clear from context. Here $S$ and $\theta$ define a scoring function $S_\theta : \mathcal{X} \to \mathbb{R}^+$. Since our focus is LLMs, we assume that queries $\mathcal{X}$ are strings of text tokens. The routing algorithm then works as follows:

$$R_\omega(x) = \begin{cases} M_{\mathtt{w}}(x) & \text{if } S_\theta(x) < \tau \\ M_{\mathtt{s}}(x) & \text{otherwise} \end{cases}$$

where $\omega = (S, \theta, \tau)$. We detail scoring functions in Section 4 and Appendix D; prior work has suggested linear models, light-weight LLMs, and more. Note that, consistent with this application, scoring functions are computationally cheap (as compared to $M_{\mathtt{s}}, M_{\mathtt{w}}$). Deployments calibrate $\tau$ to limit the fraction of queries routed to the strong model $M_{\mathtt{s}}$, giving rise to the type of control plane integrity policy discussed in Section 2.1.

## C  Gadget Examples

Below are a few examples of the optimized confounder gadgets $c$ produced by our method for different routing algorithms:

- $R_{SW}$:

> legationbelongs967reglo'hui
> (DictionaryizedNameantal bidi.numberOf

> demanding_RESULTS! XVI-ioibili Written Hormchriftlatable

- $R_{MF}$:

> 041 personne.nama tomeTN Exact relação tinha seis.Description

> NUMAGEMENTMedian157_mut-camerauacao nije Hir Ter

- $R_{CLS}$:

> contiene DEM USERNAME fourteen chees781 consultants200 inici DOJ

> 571:
> Ord:nth Norwegian Mercer_docs Abr226_METADATA

- $R_{LLM}$:

> dated:frameifyumi345 Kurdasciiuzeiphertext

> Midnightexecution431!784 below1 unwrap : / n / n

## D Experimental Setup

In this section we provide additional details regarding our experimental setup for the open-source routers evaluations, described in Section 4. All settings are summarized in Figure 3.

**Target routers.** As discussed in Section 4, we focus our evaluation on the prescriptive routing algorithms by Ong et al. (2024). These routers were trained in a supervised fashion using a set of reference (training) queries whose performance score on each of the considered models is known. The scores were computed from a collection of human pairwise rankings of model answers for each of the queries. There is no reason to believe a non-learning-based (e.g., rule-based) routing algorithm would be more adversarially robust. We now outline the four routing methods (see Ong et al. (2024) for full implementation details).

*Similarity-weighted ranking:* The first method is based on the Bradley-Terry (BT) model (Bradley & Terry, 1952). For a given user query, this model derives a function to compute the probability of the weak model being preferred over the strong model. The probability-function expressions all share parameters, which are optimized to minimize the sum of cross-entropy losses over the training-set queries, where each element in the sum is weighted by the respective query's similarity with the user's query (computed as embeddings cosine similarity, with the embedding derived using OpenAI's text-embedding-3-small (OpenAI, 2024a)). We denote this method as $R_{SW}$.

*Matrix factorization:* The second method is based on matrix factorization. The training queries are used to train a bilinear function mapping a model's embedding and a query's embedding to a score corresponding to how well the model performs on the query. Routing is done by computing the score of the input query for each model, and choosing the highest-scoring model. We denote this method as $R_{MF}$.

*BERT classifier:* The third method involves fine-tuning a classifier, based on the BERT-base architecture (Devlin et al., 2019), to predict which of the two models produces a better response for the given query or whether they do equally well (a tie). The routing decision is based on the probability of the weak model providing a better response versus the strong model or the tie. We denote this method as $R_{CLS}$.

*LLM classifier:* The last method is based on asking an LLM to provide a score in the range 1–5 of how an AI expert would struggle to respond to a given query based on the query's complexity. For this, Ong et al. fine-tuned a Llama-3-8B model (Meta, 2024a) using their reference set of queries and corresponding scores. We denote this method as $R_{LLM}$.

**Evaluation datasets.** As discussed in Section 4, we evaluate our attacks using three standard LLM benchmarks as workloads: MT-Bench (Zheng et al., 2023), a dataset of 160 open-ended questions, MMLU (Hendrycks et al., 2021), a dataset of 14,042 multi-choice questions, and GSM8K (Cobbe et al., 2021), a dataset of 1,319 grade-school math problems. In this section we provide more details regarding our evaluation of these benchmarks.

We first note that Ong et al. (2024) flagged that some data points are "contaminated", i.e., they are too similar to the ones used in their training of the routers. We use these datasets without these contaminated elements, resulting in 72 MT-bench queries, 14,037 MMLU queries, and 1,307 GSM8K queries.

For MMLU and GSM8K, we will require that the LLMs respond in a predefined format so we can parse and compare the responses to ground-truth answers. To facilitate this, we prepended formatting instructions to the query, inserted as a prefix before the gadget in the case of confounded queries. We phrase this instruction as follows: "*Answer the question using*

*the format: "Answer: [A/B/C/D]. Explanation: [EXPLANATION]""* for the multi-choice queries of the MMLU benchmark, and a similar version for GSM8K. In other words, a confounded query ends up defined as $\hat{x}_i = instr\|c\|x_i$ for instruction template *instr*, confounder gadget *c*, and original query $x_i$. Thus in this case we model a scenario where the adversary only controls a part of the prompt rather than the entire prompt. See Appendix I ablations.

When assessing the quality of responses in Section 5, we use both the perplexity scores of the responses as well as the following benchmark-specific metrics:

- MT-bench: We score the responses on a scale of 1–10 using an LLM-as-a-judge methodology (Zheng et al., 2023). We use GPT-4o (OpenAI, 2024b) as the judge and ask it to provide a score given a pair of a query and a corresponding response.

- MMLU: We parse the responses and compare the answer to the ground truth. In cases where the response did not fit any known multi-choice format, we marked the response as a mistake. We report accuracy as the percentage of responses that match the ground truth.

- GSM8K: similar to MMLU except questions are math rather than multiple choice, thus we parse the answers according to the expected format.

**Router calibration.** As mentioned in Section 4, for each workload, we must calibrate each router by setting the threshold $\tau$ to achieve some target fraction $\epsilon$ of queries routed to the strong model. For our experiments, we set $\epsilon = 0.5$. This reflects an application developer that seeks to control for costs, even if it may mean sacrificing some performance for some workloads. We evaluate other values of $\epsilon$ in Appendix E. Note that the calibration process we use is agnostic to the underlying LLM pair. We therefore must define 12 distinct thresholds, one for each router, dataset pair. We now provide more details regarding the calibration process.

To calibrate for MT-bench, we use the Chatbot Arena (Chiang et al., 2024) dataset as the calibration set, computing the threshold using the 55 K queries for which Ong et al. precomputed the scoring function outputs. To calibrate for MMLU and GSM8K, we select 1,000 queries uniformly at random and uses these to set thresholds. Looking ahead, we do not use these queries during evaluation of the attacks.

Note that it important that the distribution of calibration queries be similar to the distribution of the target workload (and, in our experiments, the test queries). We observed that the Chatbot Arena-based threshold did not transfer well to MMLU and GSM8K, resulting in the majority of queries ($\approx 98\%$) routed to the strong model.

## E Effect of stricter router thresholds

As mentioned in Section 4 and Appendix D, in all of our experiments we set $\epsilon = 0.5$. Figure 5 shows that even when using more restrictive, i.e. lower, thresholds that route fewer queries to the strong model, our attack can still reroute a significant number of queries to the strong model.

## F Runtime and convergence

Figure 6 shows the convergence rates for 10 different gadgets, against different routing algorithms. The overall average number of iterations before convergence is 58. Generation against $R_{SW}$ converges the fastest (50 iterations on average), $R_{MF}$ the slowest (66 iterations on average). Interestingly, the score of $R_{SW}$ does not increase much during optimization but is still sufficient for a successful attack.

Runtime varies significantly when generating gadgets against different routing methods. On a machine with one A40 GPU, 4 CPUs, and 180G RAM, a single iteration takes 36.9 s, 8.4 s, 0.8 s, and 6.9 s for the $R_{SW}$, $R_{MF}$, $R_{CLS}$, and $R_{LLM}$ routers, respectively. On average, it

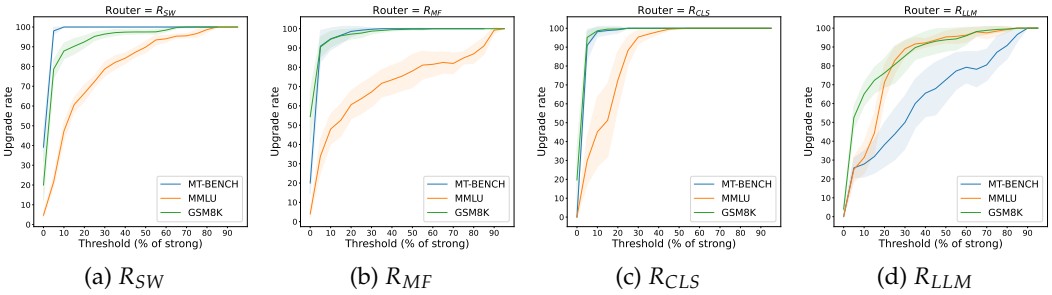

(a) $R_{SW}$    (b) $R_{MF}$    (c) $R_{CLS}$    (d) $R_{LLM}$

Figure 5: Average upgrade rate as a function of the threshold $\epsilon$. Lower thresholds indicate a stricter setting, where only a small number of queries are intended for the strong model. 50% represents the threshold used for our main results. Even in the strict setting, our attack consistently achieves a significant upgrade rate across all evaluated routers and benchmarks.

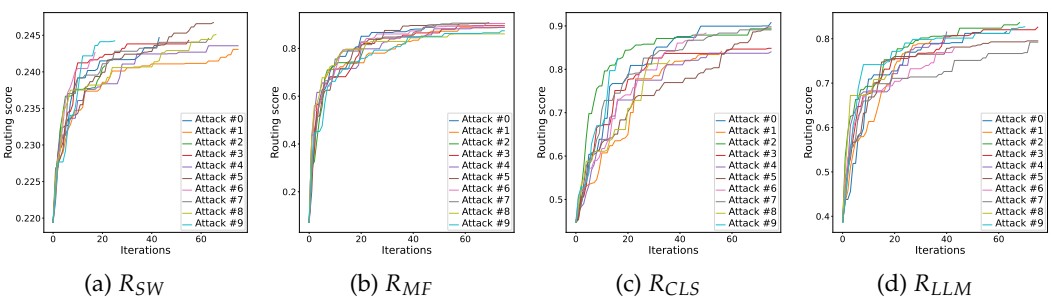

(a) $R_{SW}$    (b) $R_{MF}$    (c) $R_{CLS}$    (d) $R_{LLM}$

Figure 6: Convergence of gadget generation against different routing algorithms.

takes around 31 minutes to generate a gadget for the $R_{SW}$ router, 9 minutes for $R_{MF}$, 50s for $R_{CLS}$, and 6 minutes for $R_{LLM}$.

# G  Quality of attack responses

In Section 5 we evaluate the effectiveness of our attack in the white and black box settings. As previously discussed, a successful rerouting attack must not only reroute originally weak queries to the strong model, it must do so while preserving or even improving the quality of the LLM responses. Due to space limitations, we provide additional details and discussions over the quality evaluation in this section.

As a first measure of response quality, we compare the perplexity scores for unmodified responses and confounded query responses. Text perplexity (Jelinek, 1980) is a well-known method for approximating "naturalness" of text sequences. Perplexity can be computed using an LLM, we use GPT-2 (Radford et al., 2019) for this purpose as it is a standard choice(Alon & Kamfonas, 2023; Zhang et al., 2024a);[2] Table 2 shows the results. As can be seen, adding the confounder gadget to queries does not significantly change response perplexity. To the extent that it does, it usually somewhat decreases response perplexity, i.e., makes it more "natural". That said, perplexity is a coarse measure of "naturalness," and it does not measure whether the response is correct. In particular, responses of strong and weak LLMs tend to have similar perplexities. We further discuss this issue in Appendix K.

---

[2]A few responses had abnormally high perplexity values ($> 100$), which we found do not correlate with quality, but these variations disproportionately contribute to the average. We thus filter out such high-perplexity responses as outliers in both benign and attack settings. We provide examples and the number of filtered responses in Appendix K.

| | MT-Bench | | MMLU | | GSM8K | |
|---|---|---|---|---|---|---|
| | Original | Confounded | Original | Confounded | Original | Confounded |
| $R_{SW}$ | 10.0 | $8.7 \pm 0.3$ | 19.5 | $20.1 \pm 0.1$ | 14.5 | $15.1 \pm 0.3$ |
| $R_{MF}$ | 10.0 | $8.6 \pm 0.4$ | 19.5 | $20.2 \pm 0.2$ | 14.5 | $15.2 \pm 0.1$ |
| $R_{CLS}$ | 10.0 | $8.2 \pm 0.5$ | 19.5 | $20.3 \pm 0.2$ | 14.5 | $15.0 \pm 0.2$ |
| $R_{LLM}$ | 10.0 | $10.1 \pm 0.5$ | 19.5 | $19.7 \pm 0.1$ | 14.5 | $15.0 \pm 0.3$ |

Table 6: Average perplexity of responses to the original and confounded queries, in the white-box setting, using only the strong model of LLM pair 1.

| | MT-Bench | | MMLU | | GSM8K | |
|---|---|---|---|---|---|---|
| | Original | Confounded | Original | Confounded | Original | Confounded |
| $R_{SW}$ | 8.5 | $8.3 \pm 0.0$ | 66 | $66 \pm 0$ | 57 | $65 \pm 1$ |
| $R_{MF}$ | 8.5 | $8.3 \pm 0.1$ | 66 | $66 \pm 0$ | 57 | $67 \pm 1$ |
| $R_{CLS}$ | 8.5 | $8.4 \pm 0.1$ | 66 | $66 \pm 1$ | 57 | $63 \pm 1$ |
| $R_{LLM}$ | 8.5 | $8.3 \pm 0.1$ | 66 | $66 \pm 0$ | 57 | $65 \pm 1$ |

Table 7: Average benchmark-specific scores of responses to the original and confounded queries, in the white-box setting, using only the strong model of LLM pair 1.

We thus also evaluate using benchmark-specific metrics to assess response quality. For MT-bench, each response is ranked on a scale of 1-10 and we report the average, while for MMLU and GSM8K we report the percentage of correct responses. We provide full details of these metrics in Appendix D.

Table 3 shows that in most cases responses to the confounded queries are no worse, and in some cases even better, than responses to the original queries. We attribute the improvement on the GSM8K benchmark to the fact that the strong model performs significantly better than the weak model on this benchmark (57% vs. 33%). On the MT-bench and MMLU benchmarks, strong and weak models have comparable performance (8.5 vs. 7.6 for MT-bench and 66% vs. 64% for MMLU), thus routing does not degrade quality of responses and, consequently, the attack cannot improve it.

To further demonstrate the effect of adding the gadgets and disentangle it from the effect of rerouting to the stronger model, we compare the perplexity and benchmark scores of confounded and original queries using only the strong model. Table 6 and Table 7 show that, in most cases, the inclusion of the gadget does not significantly impact the quality of responses. Also, when manually inspecting the outputs, we observed that when the LLM produced a wrong answer for both the original and confounded queries, in most cases the answer was the same for both queries. In summary, gadgets are effective against the routers but do not substantially affect the target LLMs.

In Appendix M.1 we further demonstrate that the attack improves the quality of responses when there is a significant gap between the weak and strong LLMs by evaluating LLM pairs for which the weak model is weaker than in LLM pair 1.

LLM responses are sometimes affected by the confounder gadget. For example, an LLM responded with "I can't answer that question as it appears to be a jumbled mix of characters". Still, the response continued with "However, I can help you with the actual question you're asking," followed by the actual answer. We observed very few cases where an LLM refused to answer due to the presence of the gadget. In most cases, the response did not mention anything abnormal about the query. Intuitively, this reflects the fact that while LLMs are built to be robust to noisy inputs, the router itself is not.

## H   Black-box attack results

In addition to the evaluation of the white-box attack setting in Section 5, we evaluated the black-box transfer setting, where the attacker that does not know the algorithm used by the

| Surrogate | Target | MT-Bench | MMLU | GSM8K |
|---|---|---|---|---|
| $\hat{R}_{SW}$ | $R_{MF}$ | $99 \pm 1$ | $66 \pm 5$ | $99 \pm 1$ |
| | $R_{CLS}$ | $88 \pm 5$ | $44 \pm 11$ | $72 \pm 11$ |
| | $R_{LLM}$ | $45 \pm 5$ | $81 \pm 3$ | $63 \pm 4$ |
| $\hat{R}_{MF}$ | $R_{SW}$ | $100 \pm 0$ | $82 \pm 4$ | $92 \pm 2$ |
| | $R_{CLS}$ | $96 \pm 2$ | $56 \pm 7$ | $88 \pm 3$ |
| | $R_{LLM}$ | $39 \pm 3$ | $74 \pm 2$ | $62 \pm 4$ |
| $\hat{R}_{CLS}$ | $R_{SW}$ | $100 \pm 0$ | $64 \pm 6$ | $76 \pm 6$ |
| | $R_{MF}$ | $79 \pm 9$ | $16 \pm 7$ | $60 \pm 9$ |
| | $R_{LLM}$ | $51 \pm 5$ | $80 \pm 5$ | $65 \pm 8$ |
| $\hat{R}_{LLM}$ | $R_{SW}$ | $100 \pm 0$ | $53 \pm 4$ | $60 \pm 8$ |
| | $R_{MF}$ | $83 \pm 5$ | $20 \pm 5$ | $70 \pm 7$ |
| | $R_{CLS}$ | $85 \pm 7$ | $46 \pm 11$ | $73 \pm 10$ |

Table 8: Average upgrade rates for our attack in the black-box (transfer) setting. The average downgrade rate (i.e., strong-to-weak rerouting) is 1.2% across all routers.

| Surrogate | Target | MT-Bench | | MMLU | | GSM8K | |
|---|---|---|---|---|---|---|---|
| | | PPL | Bench | PPL | Bench | PPL | Bench |
| $\hat{R}_{SW}$ | $R_{MF}$ | 0.4 | −0.1 | 0.1 | −0.1 | 1.9 | 14.9 |
| | $R_{CLS}$ | 0.8 | −0.1 | 0.8 | 0.3 | 1.7 | 9.6 |
| | $R_{LLM}$ | 0.6 | 0.0 | 1.1 | −0.2 | 0.6 | 15.2 |
| $\hat{R}_{MF}$ | $R_{SW}$ | 1.4 | −0.1 | 0.2 | 4.8 | 1.6 | 18.6 |
| | $R_{CLS}$ | 0.7 | −0.1 | 0.2 | 1.0 | 1.7 | 13.8 |
| | $R_{LLM}$ | 0.3 | 0.0 | 1.1 | 0.5 | 0.2 | 14.7 |
| $\hat{R}_{CLS}$ | $R_{SW}$ | 1.7 | −0.1 | 0.3 | 2.5 | 1.7 | 13.4 |
| | $R_{MF}$ | 0.3 | 0.0 | 0.8 | −1.3 | 1.0 | 6.8 |
| | $R_{LLM}$ | 0.7 | 0.1 | 0.9 | −0.8 | 0.4 | 12.6 |
| $\hat{R}_{LLM}$ | $R_{SW}$ | 0.8 | −0.2 | 1.3 | 2.6 | 1.3 | 13.6 |
| | $R_{MF}$ | −0.6 | −0.1 | 1.2 | −0.9 | 1.3 | 11.3 |
| | $R_{CLS}$ | 0.0 | −0.2 | 0.9 | 0.3 | 1.7 | 10.4 |

Table 9: Differences between average perplexity ("PPL") and benchmark-specific scores ("Bench") of responses to the original and confounded queries, in the black-box setting and for LLM pair 1. Positive values indicate a lower average perplexity (more natural) and higher average score of responses to the confounded queries; higher values are better for the attacker. Standard errors were omitted for readability but are 0.2 on average for the perplexity evaluation and $0.1, 0.8,$ and $1.8$ for MT-bench, MMLU and GSM8K, respectively, for the benchmark-specific evaluation.

target router but has access to a surrogate router to mount a transferability-based attack. Due to space limitations, we only briefly mentioned the results of this evaluation in Section 5 and provide the full results here.

Table 8 shows the results for all combinations of surrogate (denoted by $\hat{R}$) and target routers. Upgrade rates are lower than in the white-box setting, but still high, i.e., the attack definitely transfers. This is also evident by the successful transferability to the commercial routers presented in Section 6. Downgrade rate is 1.2% on average and 0% in most cases.

Table 9 shows that the black-box attack does not increase the average perplexity for LLM pair 1. Table 9 also shows that the attack does not decrease benchmark-specific scores, other than a few small decreases for MMLU queries. For GSM8K, like in the white-box setting, we see an improvement with our attack as the performance gap is larger between weak and strong models. This effect is further demonstrated in Appendix M.1. In summary, confounding affects only the routing, not the quality of responses, even in the black-box setting.

|  |  | MT-Bench | MMLU | GSM8K |
|---|---|---|---|---|
| $R_{SW}$ | Prefix | $100 \pm 0$ | $90 \pm 1$ | $98 \pm 0$ |
|  | Suffix | $100 \pm 0$ | $82 \pm 2$ | $94 \pm 1$ |
| $R_{MF}$ | Prefix | $100 \pm 0$ | $78 \pm 4$ | $100 \pm 0$ |
|  | Suffix | $100 \pm 0$ | $63 \pm 3$ | $100 \pm 0$ |
| $R_{CLS}$ | Prefix | $100 \pm 0$ | $100 \pm 0$ | $100 \pm 0$ |
|  | Suffix | $100 \pm 0$ | $93 \pm 1$ | $100 \pm 0$ |
| $R_{LLM}$ | Prefix | $73 \pm 5$ | $95 \pm 1$ | $100 \pm 0$ |
|  | Suffix | $84 \pm 4$ | $93 \pm 1$ | $94 \pm 3$ |

Table 10: Average upgrade rates for different ways of adding the gadget to queries, in the white-box setting. Results are similar in both methods, with a slight preference to the prefix approach.

# I  Ablation Study

In this section, we evaluate the effect of different hyperparameters and design choices (in the white-box setting).

**Prefix vs. suffix.**  As described in Section 3, we prepend the confounder gadget to the query. An alternative is to append it. This is straightforward for MT-bench and GSM8K, but MMLU consists of multi-choice questions followed by a list of possible answers, and the term "Answer:". We insert the gadget at the end of the question text and before the possible answers. If we append it at the very end, after "Answer:", the LLM assumes the query was answered and in many cases does not generate any output at all.

Table 10 shows that average upgrade rates are similar regardless of whether the gadget was inserted as a prefix or a suffix. For MMLU, prefix works better. The downgrade rate is 0% in all cases.

As mentioned in Appendix D, to encourage the LLMs to follow the specific format in their responses (so they can be parsed and compared with the ground-truth answers), we add a short prefix to the MMLU and GSM8K queries that instructs the model how to respond. We add this instruction after modifying the queries with the confounder gadget, i.e. the instruction is prepended to the gadget.

An alternative to insert the instruction after the gadget but before the query, however we observed this to slighly underperform its counterpart. In the white-box setting we observe a slight decrease in the average (across all four routers) upgrade rate from 91% to 89% for the MMLU benchmark, and from 98% to 91% for the GSM8K benchmark. In the black-box setting, the average upgrade rate on MMLU reduces from 57% to 49% and on GSM8K from 73% to 64%.

**Token sampling method.**  When generating the confounder gadget (see Section 3), we iteratively replace tokens with the goal of maximizing the routing algorithm's score for the gadget. Candidate replacement tokens are chosen uniformly at random. An alternative is to choose candidates based on their probability of appearing in natural text. To evaluate this method, we compute token probabilities by parsing and tokenizing the wikitext-103-raw-v1 dataset (Merity et al., 2016).

Table 11 shows that in most cases uniform sampling of replacement tokens yields better upgrade rates. We conjecture that uniform sampling produces more unnatural text, confusing the router. For example, for the $R_{SW}$ routing algorithm, uniform sampling produces the following gadget: *"legationbelongs967reglo'hui(DictionaryizedNameantal bidi.numberOf"*,

| | | MT-Bench | MMLU | GSM8K |
|---|---|---|---|---|
| $R_{SW}$ | Uni. | $100 \pm 0$ | $90 \pm 1$ | $98 \pm 0$ |
| | Nat. | $100 \pm 0$ | $77 \pm 2$ | $88 \pm 2$ |
| $R_{MF}$ | Uni. | $100 \pm 0$ | $78 \pm 4$ | $100 \pm 0$ |
| | Nat. | $97 \pm 2$ | $41 \pm 3$ | $92 \pm 3$ |
| $R_{CLS}$ | Uni. | $100 \pm 0$ | $100 \pm 0$ | $100 \pm 0$ |
| | Nat. | $100 \pm 0$ | $96 \pm 2$ | $100 \pm 0$ |
| $R_{LLM}$ | Uni. | $73 \pm 5$ | $95 \pm 1$ | $94 \pm 3$ |
| | Nat. | $70 \pm 5$ | $87 \pm 4$ | $83 \pm 9$ |

Table 11: Average upgrade rates for different ways of sampling candidate tokens during gadget generation - uniformly ("Uni") or based on their probability of appearing in natural text ("Nat") , in the white-box setting. Uniformly sampling the tokens yields better upgrade rates in most cases.

whereas sampling according to natural probabilities produces "*total occurred According number Letar final Bab named remainder*".

**Number of tokens in the gadget.** In our main evaluation, the gadgets are composed of $n = 10$ tokens. We evaluate the effect of using less ($n = 5$) or more ($n = 20$ or $n = 50$) tokens. We observed that 5 tokens were insufficient to make changes to the routing algorithm's score and thus we were not able to optimize the gadget in this setting. As for 20 tokens, we observe a a small improvement in the white-box setting, increase the average upgrade rate from 93.9% to 95.8%, and a bigger improvement in the black-box setting, increase the average upgrade rate from 70.2% to 81.3%. Using 50 tokens further increases the upgrade rates, to 98.2% in the white-box setting and 84.2% in the black box setting. The average convergence rate increases as well, from 60 iterations for 10 tokens, to 70 for 20 tokens, and 100 for 50 tokens. Overall this evaluation suggests that our rerouting attack can be even further improved by using longer gadgets, however it is important to be careful not to make them too long to the point that they might degrade the performance of the underlying LLM.

## J  Optimization-Free Gadget Generation

We evaluate optimization-free alternatives to our black-box optimization method for generating confounder gadgets.

**Fixed gadget.** A simple way to create a gadget without resorting to optimization is to repeat $n$ tokens. We use ! as the initialization token, so the gadget in this case is !!!!!!!!!!. Another possibility is to select $n$ tokens uniformly at random. Table 12 shows the upgrade rates for both options, were in the latter setting we repeat the process 10 times and report the average result and the standard error. While they are non-negligible, especially for the randomly sampled gadgets, they significantly underperform the upgrade rates reported in Table 1 for optimized gadgets.

**Instruction injection.** Prompt injection is a known attack on LLMs (Perez & Ribeiro, 2022; Toyer et al., 2023), thus we consider a gadget consisting of a direct instruction to the router to treat the query as a complex one and obtain a high-quality response.

We evaluated 4 differently phrased instructions: two created manually and two generated by, respectively, Gemini (Team et al., 2023) and GPT-4o (OpenAI, 2024b), denoted as "ours-1", "ours-2", "Gemini", and "GPT".

Table 13 reports the results. This method works well in a few cases but poorly in most. This highlights the difference between attacking LLMs and attacking LLM routers.

| | gadget | MT-Bench | MMLU | GSM8K |
|---|---|---|---|---|
| $R_{SW}$ | Init | 7 | 21 | 21 |
| | Random | $97 \pm 2$ | $49 \pm 5$ | $58 \pm 8$ |
| $R_{MF}$ | Init | 3 | 4 | 20 |
| | Random | $37 \pm 8$ | $6 \pm 3$ | $34 \pm 8$ |
| $R_{CLS}$ | Init | 8 | 0 | 0 |
| | Random | $62 \pm 10$ | $14 \pm 7$ | $37 \pm 9$ |
| $R_{LLM}$ | Init | 3 | 13 | 9 |
| | Random | $38 \pm 4$ | $68 \pm 5$ | $41 \pm 7$ |

Table 12: Average upgrade rates when the gadget is not optimized and is either defined to be the the initial set of tokens or a set of uniformly sampled tokens. The optimization-based approach outperforms these optimization-free approaches.

| | intro type | MT-Bench | | MMLU | | GSM8K | |
|---|---|---|---|---|---|---|---|
| | | Up. | Down. | Up. | Down. | Up. | Down. |
| $R_{SW}$ | Ours-1 | 100 | 0 | 28 | 0 | 4 | 46 |
| | Ours-2 | 100 | 0 | 32 | 0 | 6 | 63 |
| | Gemini | 100 | 0 | 35 | 0 | 4 | 56 |
| | GPT | 100 | 0 | 54 | 0 | 4 | 77 |
| $R_{MF}$ | Ours-1 | 0 | 31 | 0 | 57 | 0 | 100 |
| | Ours-2 | 0 | 60 | 0 | 66 | 0 | 100 |
| | Gemini | 0 | 50 | 0 | 60 | 0 | 100 |
| | GPT | 0 | 48 | 0 | 51 | 0 | 100 |
| $R_{CLS}$ | Ours-1 | 33 | 8 | 2 | 47 | 0 | 77 |
| | Ours-2 | 75 | 0 | 19 | 26 | 16 | 43 |
| | Gemini | 100 | 0 | 100 | 0 | 98 | 0 |
| | GPT | 46 | 2 | 0 | 66 | 0 | 95 |
| $R_{LLM}$ | Ours-1 | 26 | 7 | 0 | 42 | 4 | 36 |
| | Ours-2 | 35 | 5 | 0 | 42 | 2 | 43 |
| | Gemini | 55 | 0 | 21 | 21 | 9 | 9 |
| | GPT | 19 | 7 | 26 | 23 | 6 | 25 |

Table 13: Average upgrade and downgrade rates of gadgets containing injected instructions to the router. This method significantly underperforms the optimization-based approach in most cases.

## K   Perplexity Issues

In Section 4, we use perplexity as one of the metrics for evaluating the effect of our attack on generated responses. In general, perplexity measures "naturalness" of the text rather than its quality. Figure 7 shows the distribution of perplexity values of the clean responses generated by weak and strong models, as well as the ROCAUC scores. Perplexity of responses is similar for the weak and strong models, with ROCAUC scores ranging between 0.38 to 0.47.

As mentioned in Section 4, throughout our evaluations we filter out responses with perplexity values higher than 100. This is due to a handful of valid responses that have abnormally high perplexity. For example, for the query:

> Suppose you are a mathematician and poet. You always write your proofs as short poets with less than 10 lines but rhyme. Prove the square root of 2 is irrational number.

The weak model responds with:

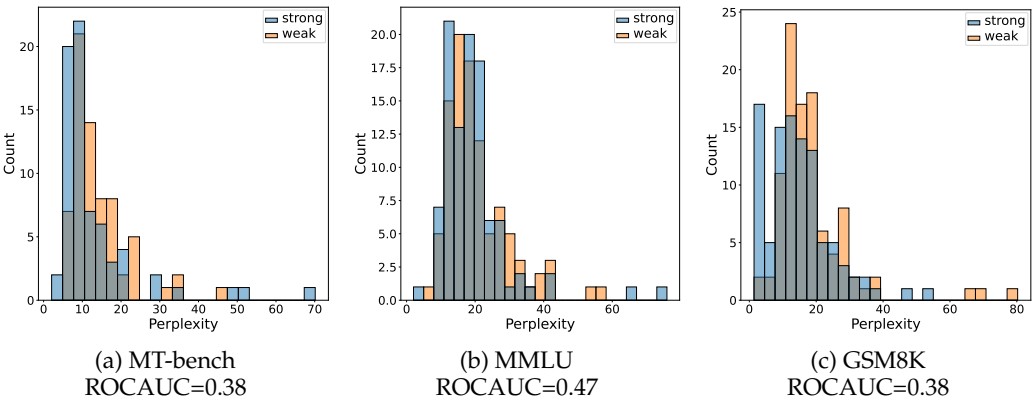

(a) MT-bench
ROCAUC=0.38

(b) MMLU
ROCAUC=0.47

(c) GSM8K
ROCAUC=0.38

Figure 7: Histograms of the perplexity values of clean responses generated by the strong and weak models.

| | MT-Bench | | MMLU | | GSM8K | |
|---|---|---|---|---|---|---|
| | Up. | Down. | Up. | Down. | Up. | Down. |
| $R_{SW}$ | $0 \pm 0$ | $24 \pm 2$ | $8 \pm 3$ | $9 \pm 2$ | $4 \pm 2$ | $48 \pm 9$ |
| $R_{MF}$ | $0 \pm 0$ | $67 \pm 6$ | $0 \pm 0$ | $77 \pm 7$ | $1 \pm 1$ | $78 \pm 11$ |
| $R_{CLS}$ | $0 \pm 0$ | $29 \pm 3$ | $0 \pm 0$ | $50 \pm 4$ | $0 \pm 0$ | $80 \pm 4$ |
| $R_{LLM}$ | $24 \pm 3$ | $1 \pm 0$ | $55 \pm 4$ | $5 \pm 1$ | $21 \pm 4$ | $4 \pm 2$ |

Table 14: Upgrade and downgrade rates for the downgrading variant of our rerouting attack, where the goal is to reroute queries to the weak model (white-box).

> In squares, two seeks a home, Two whole numbers, never roam. If it did, in pairs combined, A different square would it find. But take root, two's square, or four, Still leaves a remainder's roar. The square root of two's no child of two, Its irrational nature ever true.

which has a high perplexity value of 166. We also observed a few responses containing lists, code, or equations that result in abnormally high perplexity values. In total, we filtered 8 responses for the MT-bench dataset, 1 for MMLU, and 8 for GSM8K. We stress that this is the total across *all* evaluated settings (i.e., for all routers in the white-box setting and all surrogate-target router combinations in the black-box setting). In many settings, no responses were filtered. The maximum amount of filtered responses in a specific setting was 5.

## L   Rerouting to the Weak Model

In this section we evaluate the generality of our attack and show that generation of confounder gadgets can be optimized for the opposite objective from what we consider so far: reroute queries to the weak model. For this, we repeat the same optimization process as in Section 3 but *minimize* the router's score. Table 14 shows the upgrade and downgrade rates for this variant of the attack, in the white-box setting. In most cases we see a significant downgrade rate and a minimal upgrade rate, meaning that most of the modified queries were routed to the weak model. One notable exception is the LLM-based router $R_{LLM}$, for which the attack does not work well. Future work will be needed to explore improving confounder generation for this setting further.

| | MT-Bench | | MMLU | | GSM8K | |
|---|---|---|---|---|---|---|
| | Original | Confounded | Original | Confounded | Original | Confounded |
| $R_{SW}$ | 9.2 | $9.2 \pm 0.0$ | 76 | $84 \pm 1$ | 62 | $86 \pm 0$ |
| $R_{MF}$ | 9.1 | $9.3 \pm 0.0$ | 76 | $81 \pm 0$ | 65 | $88 \pm 1$ |
| $R_{CLS}$ | 9.2 | $9.1 \pm 0.0$ | 76 | $84 \pm 0$ | 68 | $90 \pm 2$ |
| $R_{LLM}$ | 8.9 | $9.1 \pm 0.1$ | 78 | $84 \pm 1$ | 66 | $85 \pm 2$ |

Table 15: Benchmark-specific average scores of responses to the original and confounded queries with GPT-4-1106-preview as the strong model (LLM pair 4), in the white-box setting. Results demonstrate a higher increase in performance with respect to the LLM pair 1 setting, due to the larger performance gap between the models.

## M   Results for Other LLM Pairs

As discussed in Section 4, for allowing our evaluation to scale, we use as the strong model $M_{\mathrm{s}}$ the open-sourced Llama-3.1-8B (Meta, 2024b) and as $M_{\mathrm{w}}$ the 4-bit quantized version of Mixtral 8x7B, denoted in our work as LLM pair 1. In this section we extend our evaluation to other LLM pairs.

We begin by evaluating the case where the weak model produces much worse responses than the strong model. We define LLM pair 2 as the strong model plus Mistral-7B-Instruct-v0.3 (Jiang et al., 2023a) and LLM pair 3 as the strong model plus Llama-2-7B-chat-hf (Touvron et al., 2023). The weaker models in pairs 2 and 3 were chosen to represent smaller (Mistral 7B) and older-generation (Llama-2) models: according to the Chatbot Arena leaderboard (Face, 2024; Chiang et al., 2024), Llama-3.1-8B is ranked in the 58th place, Mixtral 8x7B at the 88th place, Mistral-7B at the 108th place, and Llama-2-7B at the 125th place.

We additionally perform some smaller-scale evaluations using the LLM pair originally used by Ong et al. (2024), i.e., GPT-4-1106-preview (Achiam et al., 2023) as the strong model and Mixtral 8x7B (Jiang et al., 2024) as the weak model. We refer to this setting as LLM pair 4.

Figure 3 in the appendix shows the strong-weak pairs in our experiments.

### M.1   Results for LLM Pairs 2 and 3

The discussion over quality of attack responses in Section 5 shows that for the white-box setting, in most cases, responses to the confounded queries are no worse, and in some cases even better, than responses to the original queries. To further demonstrate that the attack improves the quality of responses when there is a significant gap between the weak and strong LLMs, we perform an additional evaluation with Mistral-7B-Instruct-v0.3 (Jiang et al., 2023a) and Llama-2-7B-chat-hf (Touvron et al., 2023) as the weak LLMs (LLM pairs 2 and 3). Mistral-7B achieves 7.4, 57%, and 25% on MT-bench, MMLU, and GSM8K, respectively. Llama-2-7B achieves 6.4, 44%, and 21%. Table 16 shows that the rerouting attack improves the quality of responses when either of these LLMs is the weak model, and in particular for the weaker Llama-2-7B model.

Table 17 shows similar behaviour for the black box setting, with the benchmark-specific scores improving when the weak model is significantly weaker than the strong model, i.e., LLM pairs 2 and 3.

### M.2   Results for LLM Pair 4

As discussed in Section 4, we replace the strong model that was used by Ong et al. (2024), GPT-4-1106-preview (rank 28 in the Chatbot Arena leaderboard (Face, 2024; Chiang et al., 2024)), with the open-sourced Llama-3.1-8B (rank 58) to reduce the costs of our extensive set of evaluations. In this section we perform a smaller-scale evaluation of the quality-enhancing attack performance when using GPT as the strong model, i.e., LLM pair 4. We evaluate this setting using three of the $n = 10$ confounder gadgets for each router.

|  | MT-Bench | | MMLU | | GSM8K | |
|---|---|---|---|---|---|---|
|  | Orig. | Conf. | Orig. | Conf. | Orig. | Conf. |
| | | | LLM pair 2 | | | |
| $R_{SW}$ | 8.5 | $8.3 \pm 0.0$ | 55 | $64 \pm 1$ | 46 | $64 \pm 1$ |
| $R_{MF}$ | 8.4 | $8.3 \pm 0.1$ | 63 | $64 \pm 0$ | 51 | $67 \pm 1$ |
| $R_{CLS}$ | 8.4 | $8.4 \pm 0.1$ | 58 | $66 \pm 1$ | 49 | $63 \pm 1$ |
| $R_{LLM}$ | 8.4 | $8.3 \pm 0.1$ | 62 | $66 \pm 0$ | 38 | $63 \pm 2$ |
| | | | LLM pair 3 | | | |
| $R_{SW}$ | 8.4 | $8.3 \pm 0.0$ | 51 | $64 \pm 1$ | 40 | $64 \pm 1$ |
| $R_{MF}$ | 8.1 | $8.3 \pm 0.1$ | 57 | $63 \pm 1$ | 44 | $67 \pm 1$ |
| $R_{CLS}$ | 8.3 | $8.4 \pm 0.1$ | 52 | $66 \pm 1$ | 45 | $63 \pm 1$ |
| $R_{LLM}$ | 8.1 | $8.2 \pm 0.1$ | 59 | $66 \pm 1$ | 37 | $64 \pm 1$ |

Table 16: Average benchmark-specific scores of responses to the original and confounded queries with Mistral-7B-Instruct-v0.3 (LLM pair 2) or Llama-2-7B-chat-hf (LLM pair 3) as the weak model, in the white-box setting. Results further emphasize that the rerouting attack improves quality of responses when there is a significant gap between the weak and strong LLMs.

| Surrogate | Target | LLM pair 2 | | | LLM pair 3 | | |
|---|---|---|---|---|---|---|---|
| | | MT-Bench | MMLU | GSM8K | MT-Bench | MMLU | GSM8K |
| $\hat{R}_{SW}$ | $R_{MF}$ | $-0.1$ | 1.6 | 13.6 | 0.2 | 5.0 | 20.5 |
| | $R_{CLS}$ | $-0.1$ | 4.0 | 8.7 | 0.0 | 6.8 | 13.4 |
| | $R_{LLM}$ | $-0.1$ | 4.2 | 18.5 | 0.1 | 5.8 | 20.9 |
| $\hat{R}_{MF}$ | $R_{SW}$ | $-0.2$ | 7.9 | 18.9 | $-0.1$ | 11.3 | 24.3 |
| | $R_{CLS}$ | $-0.2$ | 5.0 | 14.4 | $-0.1$ | 9.1 | 18.6 |
| | $R_{LLM}$ | $-0.2$ | 4.4 | 18.3 | 0.0 | 4.7 | 21.6 |
| $\hat{R}_{CLS}$ | $R_{SW}$ | $-0.1$ | 5.0 | 13.1 | 0.0 | 8.1 | 17.9 |
| | $R_{MF}$ | $-0.1$ | $-2.9$ | 4.0 | 0.2 | $-3.7$ | 11.2 |
| | $R_{LLM}$ | 0.0 | 3.2 | 15.5 | 0.2 | 4.8 | 18.9 |
| $\hat{R}_{LLM}$ | $R_{SW}$ | $-0.2$ | 5.2 | 11.3 | $-0.1$ | 7.8 | 16.7 |
| | $R_{MF}$ | $-0.2$ | $-0.9$ | 8.4 | 0.1 | 0.1 | 15.2 |
| | $R_{CLS}$ | $-0.2$ | 3.8 | 10.8 | $-0.1$ | 7.2 | 14.2 |

Table 17: Differences between average benchmark specific scores of responses to the original and confounded queries, when the confounder gadget was generated for a different surrogate router than the target (black-box setting) for LLM pairs 2 and 3. Positive values indicate a higher average score for responses to the confounded queries; higher values are better for the attacker. Results are averaged across gadgets. Standard errors were omitted for readability and are on average $0.1, 0.8$, and $1.8$ for MT-bench, MMLU and GSM8K, respectively. Aligned with the white-box setting, results show almost no decrease in performance, and improvement when there is a performance gap for the LLM pair.

Table 15 shows the results across benchmarks in the white-box setting. Compared to the pair 1 setting (Table 3), the attack results in a higher increase in benchmark performance. This further demonstrates higher attack effect on response quality when the performance gap between the weak and strong models is higher.

# N  Query-specific Gadgets

By default, our gadget generation method is query-independent and the same gadget can be used to reroute any query. An adversary with more resources may instead generate a dedicated gadget for each query. Our confounder gadget approach, provided in Section 3, extends to this setting readily by replacing $S_\theta(c)$ in Eq. 3 with $S_\theta(c\|x_i)$.

|  | MT-Bench | MMLU | GSM8K |
|---|---|---|---|
| $R_{SW}$ | 100 | 100 | 100 |
| $R_{MF}$ | 100 | 96 | 100 |
| $R_{CLS}$ | 100 | 100 | 100 |
| $R_{LLM}$ | 100 | 100 | 100 |

Table 18: Upgrade rates for query-specific gadgets, in the white-box setting. Results are nearly perfect, i.e. nearly all confounded queries are routed to the strong model.

| Surrogate | Target | MT-Bench | MMLU | GSM8K |
|---|---|---|---|---|
| $\hat{R}_{SW}$ | $R_{MF}$ | 100 | 96 | 100 |
|  | $R_{CLS}$ | 83 | 57 | 68 |
|  | $R_{LLM}$ | 71 | 89 | 74 |
| $\hat{R}_{MF}$ | $R_{SW}$ | 100 | 95 | 100 |
|  | $R_{CLS}$ | 83 | 43 | 73 |
|  | $R_{LLM}$ | 48 | 83 | 68 |
| $\hat{R}_{CLS}$ | $R_{SW}$ | 100 | 74 | 81 |
|  | $R_{MF}$ | 73 | 13 | 65 |
|  | $R_{LLM}$ | 52 | 83 | 70 |
| $\hat{R}_{LLM}$ | $R_{SW}$ | 100 | 77 | 88 |
|  | $R_{MF}$ | 67 | 11 | 54 |
|  | $R_{CLS}$ | 83 | 30 | 64 |

Table 19: Upgrade rates for query-specific gadgets, in the black-box setting. In most cases results are better than in the query-independent setting, at the cost of a more resource intensive process.

Table 18 and Table 19 show the results for the white-box and black-box settings, respectively. (Here, percentage numbers are not averaged and there is no standard error since we used a single gadget per query.) The white-box results are nearly perfect; the black-box results are often better but sometimes somewhat worse than those for query-independent gadgets. We conjecture that this is due to some level of overfitting.

## O   Adaptive Attack Against Perplexity Defense

In each iteration $t \in [T]$, we uniformly sample a target index $j \in [1, n]$ and generate a set $\mathcal{B}$ of $B + 1$ candidates as explained in Section 3. We then modify Eq. 3 such that we now find the candidate that maximizes the difference between the router's score and the perplexity constraint for the confounder:

$$c^{(t+1)} \leftarrow \underset{c \in \mathcal{B}}{\arg\max} \ \left( S_\theta(c \| x_i) - \alpha \cdot |\mathsf{PPL}(c) - \rho| \right),$$

where $\mathsf{PPL}(\cdot)$ denotes the perplexity function computed using GPT-2, the value $\rho$ denotes a target perplexity value to which we want gadgets' perplexity to be close, and the value $\alpha$ is a balancing coefficient. For the experiments below, we set $\rho$ to be the average perplexity value of 100 uniformly sampled queries[3] from the GSM8K benchmark.

Figure 8 shows the results when setting $\alpha = 0.01$, for the GSM8K benchmark and one confounder gadget generated for the $R_{SW}$ router. Modified queries can no longer be easily distinguished from normal queries by their perplexity alone. For instance, in the case of the $R_{SW}$ router, setting the threshold value at 55 as before, no confounded queries are flagged as anomalous, meaning the true-positive rate is zero. We note that there is some variability across gadgets. The average ROCAUC scores of the defense across ten gadgets with standard deviation indicated parenthetically, are 0.66 ($\pm$0.04), 0.69 ($\pm$0.02), 0.71 ($\pm$0.02), and 0.69 ($\pm$0.03) for the $R_{SW}, R_{MF}, R_{CLS}$, and $R_{LLM}$ routers, respectively.

---

[3]The perplexity calibration queries were chosen such that they do not overlap with the queries used for evaluation.

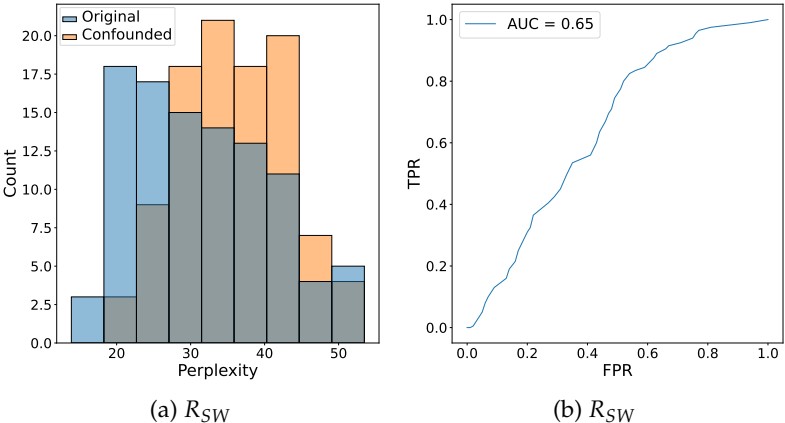

(a) $R_{SW}$             (b) $R_{SW}$

Figure 8: Perplexity values of the original and confounded queries, and the corresponding ROC curves of the defense that detects confounded queries by checking if they cross a perplexity threshold, when the confounder gadget is optimized for low perplexity, in the GSM8K benchmark and for one gadget sampled uniformly at random. Confounded queries have similar perplexity values as the original queries, and can no longer be easily distinguished based on perplexity alone.

|  | MT-Bench | | MMLU | | GSM8K | |
|---|---|---|---|---|---|---|
|  | Orig. | PPL-opt. | Orig. | PPL-opt. | Orig. | PPL-opt. |
| $R_{SW}$ | $100 \pm 0$ | $100 \pm 0$ | $90 \pm 1$ | $59 \pm 5$ | $98 \pm 0$ | $70 \pm 7$ |
| $R_{MF}$ | $100 \pm 0$ | $98 \pm 2$ | $78 \pm 4$ | $74 \pm 5$ | $100 \pm 0$ | $98 \pm 2$ |
| $R_{CLS}$ | $100 \pm 0$ | $98 \pm 1$ | $100 \pm 0$ | $66 \pm 12$ | $100 \pm 0$ | $88 \pm 6$ |
| $R_{LLM}$ | $73 \pm 5$ | $51 \pm 8$ | $95 \pm 1$ | $89 \pm 3$ | $94 \pm 3$ | $81 \pm 8$ |

Table 20: Average upgrade rates for gadgets generated without ("Orig.") and with ("PPL-opt.") low-perplexity optimization, for the balancing coefficient $\alpha = 0.01$. In some cases, optimizing for low perplexity has a negative effect on the attack success rate, however the attack can still be considered successful. A more careful choice of $\alpha$ can potentially limit the effect on the attack success.

At the same time, optimizing for low perplexity does not significantly impact the attack success rate. Table 20 compares the average upgrade rates (over $n = 10$ gadgets) of the original perplexity-agnostic optimization approach from Section 3 and the perplexity-minimizing one described above. The attack efficacy might be improvable further by adjusting $\alpha$ to find a sweet spot that avoids the defense effectively while ensuring high rerouting success rate.

We additionally compare both optimization approaches in terms of response quality. Table 21 and Table 22 compare the average perplexity values and benchmark-specific scores of responses. Results indicate that both types of gadgets have similar (minor) effect on the quality of responses.

The attack is not particularly sensitive to the choice of queries used to obtain the calibration value $\rho$. Although $\rho$ was computed using GSM8K queries, we observe similar performance when evaluating on MT-bench and MMLU, with average ROCAUC scores of 0.50 ($\pm 0.01$), 0.51 ($\pm 0.01$), 0.52 ($\pm 0$), and 0.51 ($\pm 0.01$) for MT-bench, and 0.52 ($\pm 0.03$), 0.54 ($\pm 0.02$), 0.55 ($\pm 0.01$), and 0.53 ($\pm 0.02$) for MMLU. One might also remove the calibration value altogether, instead simply minimizing the gadget's perplexity value. This can produce an "overshooting" effect, where the perplexity value is significantly *lower* than that of normal queries, thereby making it still distinguishable from standard queries.

In summary, perplexity-based filtering is not an effective defense against against rerouting.

|          | MT-Bench | | MMLU | | GSM8K | |
|----------|----------|----------|----------|----------|----------|----------|
|          | Orig.    | PPL-opt. | Orig.    | PPL-opt. | Orig.    | PPL-opt. |
| $R_{SW}$  | $12.3 \pm 0.2$ | $12.0 \pm 0.2$ | $20.1 \pm 0.1$ | $19.9 \pm 0.2$ | $15.1 \pm 0.3$ | $15.8 \pm 0.4$ |
| $R_{MF}$  | $12.3 \pm 0.2$ | $11.9 \pm 0.1$ | $20.3 \pm 0.1$ | $20.4 \pm 0.2$ | $15.2 \pm 0.3$ | $15.9 \pm 0.3$ |
| $R_{CLS}$ | $12.1 \pm 0.2$ | $12.3 \pm 0.2$ | $20.5 \pm 0.1$ | $20.0 \pm 0.2$ | $15.0 \pm 0.2$ | $15.1 \pm 0.3$ |
| $R_{LLM}$ | $12.7 \pm 0.4$ | $11.9 \pm 0.1$ | $19.6 \pm 0.1$ | $19.5 \pm 0.2$ | $15.2 \pm 0.3$ | $15.0 \pm 0.3$ |

Table 21: Average perplexity of responses to gadgets generated without ("Orig.") and with ("PPL-opt.") low-perplexity optimization, for the balancing coefficient $\alpha = 0.01$. Both methods perform similarly.

|          | MT-Bench | | MMLU | | GSM8K | |
|----------|----------|----------|----------|----------|----------|----------|
|          | Orig.    | PPL-opt. | Orig.    | PPL-opt. | Orig.    | PPL-opt. |
| $R_{SW}$  | $8.3 \pm 0.0$ | $8.3 \pm 0.1$ | $66 \pm 0$ | $63 \pm 1$ | $64 \pm 1$ | $61 \pm 2$ |
| $R_{MF}$  | $8.4 \pm 0.0$ | $8.2 \pm 0.1$ | $64 \pm 1$ | $66 \pm 1$ | $67 \pm 1$ | $63 \pm 1$ |
| $R_{CLS}$ | $8.3 \pm 0.0$ | $8.3 \pm 0.0$ | $65 \pm 0$ | $63 \pm 1$ | $63 \pm 1$ | $63 \pm 1$ |
| $R_{LLM}$ | $8.2 \pm 0.1$ | $8.3 \pm 0.0$ | $66 \pm 0$ | $67 \pm 0$ | $64 \pm 1$ | $61 \pm 2$ |

Table 22: Average benchmark-specific scores of responses to gadgets generated without ("Orig.") and with ("PPL-opt.") low-perplexity optimization, for the balancing coefficient $\alpha = 0.01$. Both methods perform similarly.

## P Extended Survey of Related Work

**Evasion attacks against ML systems.** A large body of work investigated evasion attacks against ML systems Dalvi et al. (2004); Lowd & Meek (2005); Szegedy et al. (2013), aka adversarial examples Goodfellow et al. (2015); Papernot et al. (2016; 2017), including text-only (e.g., Cho et al. (2024)) and multi-modal LLMs Dong et al. (2023). In Appendix A, we explained the differences and similarities between LLM control plane attacks and adversarial examples.

**Prompt injection against LLMs.** Prompt injection involves manipulating the prompt, i.e., input to the LLM, causing it to generate outputs that satisfy some adversarial objective Perez & Ribeiro (2022); Toyer et al. (2023). Evasion attacks discussed above can use prompt injection. For example, jailbreaking attacks bypass some safety guardrail of the target LLM, such as "do not output expletives" Liu et al. (2023); Schulhoff et al. (2023); Zou et al. (2023); Wei et al. (2023); Zhu et al. (2023); Chu et al. (2024).

Prompt injection can also extract information from or about the model, e.g., the system prompt Perez & Ribeiro (2022); Zhang et al. (2024b); Schulhoff et al. (2023), training data Nasr et al. (2023), or model parameters Carlini et al. (2024). In indirect prompt injection Greshake et al. (2023), adversaries do not directly interact with the target LLM and instead inject adversarial inputs into third-party data, which is then added to the LLM prompt (intentionally or unintentionally) by the victim application and/or its users. A related category of attacks exploits weaknesses of LLMs to compromise RAG, retrieval-augmented generation Chaudhari et al. (2024); Shafran et al. (2024).

Our attacks, too, modify LLM queries, but the objective is different: to change routing decisions while preserving or improving outputs. Future work may investigate indirect attacks that somehow trick users into forming queries that confound control planes of LLM-based applications.

**Attacks against MoE.** Mixture-of-Experts (MoE) architectures reduce inference cost by routing different tokens of the query to different "expert" sub-models Du et al. (2022); Fedus et al. (2022); Riquelme et al. (2021); Shazeer et al. (2016). This can be thought of as an internal router within a single LLM.

Hayes et al. Hayes et al. (2024) identified a vulnerability in MoE systems that can be exploited for denial of service. The connection between token-routing, single-LLM attacks and multi-LLM attacks studied in this paper can be explored in future work.

Yona et al. Yona et al. (2024) presented a side-channel attack on MoE that enables an attacker to infer other users' prompts. There may exist side-channel attacks against multi-LLM control planes that, for example, identify the underlying LLMs. These attacks would target confidentiality rather than integrity.

